# Spin-Label Electron Paramagnetic Resonance Spectroscopy Reveals Effects of Wastewater Filter Membrane Coated with Titanium Dioxide Nanoparticles on Bovine Serum Albumin

**DOI:** 10.3390/molecules28196750

**Published:** 2023-09-22

**Authors:** Krisztina Sebők-Nagy, Zoltán Kóta, András Kincses, Ákos Ferenc Fazekas, András Dér, Zsuzsanna László, Tibor Páli

**Affiliations:** 1Institute of Biophysics, Biological Research Centre Szeged, 6726 Szeged, Hungary; seboknagy.krisztina@brc.hu (K.S.-N.); kota.zoltan@brc.hu (Z.K.); kincses.andras@brc.hu (A.K.); der.andras@brc.hu (A.D.); 2Department of Biosystems Engineering, Faculty of Engineering, University of Szeged, 6725 Szeged, Hungary; fazekas@mk.u-szeged.hu (Á.F.F.); zsizsu@mk.u-szeged.hu (Z.L.)

**Keywords:** wastewater cleaning, polyvinylidene fluoride filter membrane, titanium dioxide nanoparticles, P25, serum albumin, spin-label EPR, fluorescence, dynamic light scattering

## Abstract

The accumulation of proteins in filter membranes limits the efficiency of filtering technologies for cleaning wastewater. Efforts are ongoing to coat commercial filters with different materials (such as titanium dioxide, TiO_2_) to reduce the fouling of the membrane. Beyond monitoring the desired effect of the retention of biomolecules, it is necessary to understand what the biophysical changes are in water-soluble proteins caused by their interaction with the new coated filter membranes, an aspect that has received little attention so far. Using spin-label electron paramagnetic resonance (EPR), aided with native fluorescence spectroscopy and dynamic light scattering (DLS), here, we report the changes in the structure and dynamics of bovine serum albumin (BSA) exposed to TiO_2_ (P25) nanoparticles or passing through commercial polyvinylidene fluoride (PVDF) membranes coated with the same nanoparticles. We have found that the filtering process and prolonged exposure to TiO_2_ nanoparticles had significant effects on different regions of BSA, and denaturation of the protein was not observed, neither with the TiO_2_ nanoparticles nor when passing through the TiO_2_-coated filter membranes.

## 1. Introduction

The efficiency of the filtering of biomolecules certainly depends on the effect of the surface of the filter on the biomolecules. The separation of proteins from aqueous solutions can be efficiently achieved by membrane filtration. However, membrane fouling seriously limits its application when proteins tend to adsorb to the surface of the membrane pores, hence reducing the pass-through efficiency and, ultimately, the lifespan of the membranes. An ideal filter membrane would keep proteins in the pre-filter space without losing its filtering efficiency. Until now, several new methods have been developed to produce antifouling membranes, such as membrane modification by nanoparticles (e.g., titanium dioxide, TiO_2_). Although a number of modified membranes were developed, there is only limited information available regarding whether the applied nanoparticles cause any changes in the protein structure, altering their filterability and other biophysical properties [1,2,3]. Our previous results showed that coating membranes with TiO_2_ kept the membrane structure intact; however, it surprisingly worsened the retention of bovine serum albumin (BSA) [4,5]. This phenomenon can only be partially explained by changes in the roughness and morphology of the membrane surface, and previous investigations have also raised the possibility of changes in the protein structure as well [6]. In the present study, we address precisely this aspect of filtering out proteins from wastewater. We have chosen BSA as the model protein in this work, not only because other teams [7,8,9,10,11] and we also have used it in our recent relevant studies [4,5,12], but also because it is a well-known protein, with plenty of biophysical data about its water-soluble state [13,14,15,16,17,18], that can serve as reference for comparison. Spin-label EPR spectroscopy is our main technique here because it has proven to be among the most powerful techniques in studying serum albumins [19,20,21,22]. Commercial PVDF filter membranes were modified by coating them with TiO_2_ nanoparticles (P25), and the effects of the filtration through them and the contact with TiO_2_ on the protein structure were investigated by spin-label EPR and native fluorescence spectroscopy, as well as DLS, to reveal the potential changes in the protein structure.

Apart from their diverse physiological functions, serum albumins are the major vehicles of fatty acid transport through blood plasma, as they can bind several long-chain fatty acids with high affinity [23,24,25,26]. Due to this fatty-acid-binding function of BSA [27,28,29,30,31] and because spin-label EPR spectroscopy played a crucial role in identifying and characterising the fatty-acid-binding sites of serum albumins [27,30,31,32,33], we have used a spin-labelled stearic acid analogue (5-SASL) to detect changes in the fatty acid binding of the protein in its process-relevant interactions with the membrane and the nanoparticles. Fatty acid binding to serum albumins has proven to be sensitive to the physical state of the protein (affected by, e.g., pH, ligand-induced allosteric modulation, and temperature) [25,27,28,32]. We have also used two common maleimide-type spin labels binding to unblocked Cys residues to detect changes in the dynamics of the spin-labelled Cysteines and structural changes in their vicinity [21,34,35,36,37]. According to the crystal structure [38], the labelled residue is likely to be Cys34 (in Figure 1, the blue-coloured residue in the sequence and structure [39,40]) because the others are participating in disulfide bridges [21,28].

Since BSA has fluorescent residues, it can be studied with fluorescence spectroscopy without attaching a fluorescent dye to the protein [41,42,43]. Indeed, the two Tryptophan residues of BSA (in Figure 1, magenta-coloured) are sensitive to conformational changes via the altered polarity and rotational dynamics [42,43,44,45]. Therefore, we have carried out fluorescence spectroscopic measurements for BSA in the absence and presence of TiO_2_ nanoparticles and for filtrated BSA. Since BSA may be present in different states in an aqueous solution depending on its concentration and other conditions, such as folded or denatured monomeric or multimeric, aggregated and micellar forms [10,14,22,46], we tested changes in the particle size distribution of BSA upon the above different treatments using dynamic light scattering (DLS).

## 2. Results and Discussion

### 2.1. Chemical Oxygen Demand (COD)

Ultrafiltration experiments were carried out to investigate the effect of a TiO_2_ coating on the filtration performance of composite membranes, while microfiltration was used in another series of experiments aimed at investigating the potential effect of TiO_2_ nanoparticles on BSA structure and filtration behaviour. In this case, the aim of microfiltration was to separate the TiO_2_ nanoparticles from the BSA solution. The BSA rejection of the pristine and modified membranes in the presence (BSA@PVDF/TiO_2_) or absence of TiO_2_ (BSA@PVDF) is illustrated in Table 1. 

It was found that the pristine membrane rejects more BSA than the modified one. This is a very surprising result; thus, we also checked if the higher BSA permeability of the membrane was caused by the contact of the BSA with TiO_2_ nanoparticles. The BSA was mixed with TiO_2_; then, the BSA was separated from the TiO_2_ by microfiltration, and the clean BSA solution was filtrated through the PVDF membrane (BSA(TiO_2_)@PVDF). It was found that the contact of the BSA with TiO_2_ resulted in no noticeable change in the filtration performance after it was separated from the TiO_2_. The filtration alone through cellulose acetate (CA) membrane caused an approx. 5% BSA loss from the 1.5 × 10^−5^ M solution, while the retention during the filtration through the PVDF membrane was dependent on the BSA concentration: in three-fold-diluted solutions, the retention decreases. Nevertheless, this cannot explain the decreased retention of the BSA filtrated through the TiO_2_-modified membrane; thus, in further experiments, the potential effect of the contact between the TiO_2_ and BSA on BSA structure was investigated.

### 2.2. Dynamic Light Scattering (DLS)

We tested the particle size distribution of the BSA in our samples with dynamic light scattering (DLS). Table 2 summarises our data and the literature data, along with the different states of BSA in an aqueous environment. The DLS curves for the stock (6 × 10^−5^ M) and the 100-fold-diluted BSA solution can be seen in Figure 2.

A narrow peak at around a mean particle size of 1.7 nm, and two very broad and highly overlapping peaks at 24.4 and 85–102 or 105–200 nm are present in the BSA stock solutions (Figure 2A). The 100-fold-diluted BSA solution (Figure 2B) results in the disappearance of the peaks at 1.7 nm, and the appearance of a new peak at ~11 nm can be observed, along with the strong reduction in the broad (20–1000 nm) scattering region. There are a wide range of particle sizes reported in the literature for BSA in different concentration regions and conditions as summarised in Table 2. 

Comparing our results with this set of previously reported data, we can conclude that the 11 nm peak in our 100-fold-diluted solution is likely to correspond to dimers, whereas the non-diluted BSA dispersion contains large aggregates and a very minor contribution of small (probably unfolded) monomers (considering the logarithmic scale of the x-axis) [48]. It is important to note that, in the present work, the concentration of BSA was 10 or 100 times higher (for the diluted and the stock solutions, respectively) than the 6.5–6.9 × 10^−7^ M critical micelle concentration (CMC) [16]. Therefore, the disappearance or dissolution of micelles upon their filtering and interaction with TiO_2_ can be excluded. It should be noted here that the average size of the membrane pore was estimated to be ~14 nm by Howe and Clark [49], whereas the size of the primer TiO_2_ particles is ~25.4 nm, but they form aggregates in water with a diameter of ~1 μm [50]. There is, therefore, a very low probability for the TiO_2_ particles to pass through the membrane pores; consequently, the TiO_2_ content must be very low in the filtrated BSA sample.

### 2.3. Samples for Fluorescence and EPR Spectroscopy

Regarding the industrial process of wastewater filtration, we have used four types of samples in our spectroscopic measurements: (i) 6 × 10^−5^ M BSA in water, (ii) 2 × 10^−2^ M TiO_2_ in water, (iii) BSA-TiO_2_ mixture in water where [BSA] = 6 × 10^−5^ M and [TiO_2_] = 2 × 10^−2^ M, and (iv) 6 × 10^−5^ M BSA filtrated through composite ultrafilter membranes. These samples are marked as “BSA”, “TiO_2_”, “BSA+TiO_2_”, and “filtrated BSA”, respectively, in the EPR and fluorescence spectroscopic experiments. Some samples had to be diluted 30-fold for optimal conditions in the fluorescence experiments, as mentioned below. Other than this, we did not apply any centrifugation, washing, filtering, or other separation techniques on the samples, to exclude any dilution and concentration effects. Therefore, the free (unbound) spin probe was not removed from the EPR samples. It is important to note that, due to the filtration, the protein concentration in the filtrated BSA samples was only 1.8 × 10^−5^ M—that is, 3.3-fold lower than in the BSA and BSA+TiO_2_ samples. Since BSA is sensitive to pH [27,43], which might be affected by a high concentration of TiO_2_, we have measured the pH of the above four types of samples and obtained values of 7.4, 4.9, 7.0, and 7.3 for the BSA, TiO_2_, BSA+TiO_2_, and filtrated BSA samples, respectively. (We did not buffer the pH to the same value in the different samples because it would be incompatible with the industrial wastewater filtration process.) The corresponding pH values for the 30-fold-diluted samples were 6.7, 6.1, 6.7, and 6.4, respectively. This result shows that the pH effect of the TiO_2_ is compensated for by the ~2.5-times-higher concentration of BSA.

### 2.4. Fluorescence Spectroscopy

BSA has two Tryptophan residues (Trp134 and Trp213; see Figure 1) with fluorescence emission maxima at ~348 nm in pure water [51]. The fluorescence from the Tyrosines is negligible compared to that of the Tryptophanes [42]. We have made fluorescence spectroscopic measurements on all four types of samples from their stock solutions. Table 3 contains the wavelengths of the emission maxima (λ_max_) (from Gaussian fits) of the BSA-containing samples and the (weighted) mean fluorescence emission maximum (<λ_F_>) as calculated by Equation (1) [52].
(1)$${<\lambda }_{F}>=\frac{\sum f_{\lambda }\cdot \lambda }{\sum f_{\lambda }}$$
where λ is the wavelength and f_λ_ is the emission intensity at λ.

The emission maximum (λ_max_) and the mean emission maximum (<λ_F_>) for the filtrated sample were 340.2 nm and 356.9 nm, respectively, which are red-shifted by 1.3 nm and 4.6 nm, respectively, relative to the BSA sample (with λ_max_ = 338.9 and <λ_F_> = 352.3 nm). The BSA+TiO_2_ sample could not be measured at this concentration because of strong light scattering. Therefore, the samples were diluted 30-fold to reduce the disturbing light scattering (Figure 3). Dilution alone resulted in a negligible blue shift (0.5 nm) for the BSA and red shift (0.5 nm) for the filtrated BSA samples relative to the original samples in the case of the fitted emission maximum. The mean emission maximum showed little blue shift (1.9 nm) for the filtrated BSA. The diluted BSA+TiO_2_ sample yielded an intensity maximum at 355.6 nm (<λ_F_> = 363.3 nm), which means a strong red shift relative to the BSA and filtrated BSA (Figure 3, Table 3). It should be noted that TiO_2_ did not give a measurable contribution to the fluorescence spectra (Figure 3).

In the case of Tryptophan, any negative charge in the environment of the pyrrole ring or any positive charge close to the benzene ring causes a change in the electron densities of both rings [44], hence resulting in a bathochromic shift in the fluorescence spectrum. Based on quantum mechanics (DFT) calculations on TiO_2_, it has been reported that the charge state of Ti is +3 and the oxygen is −1.5 in the molecule [53]. This means that TiO_2_ is very polar and able to induce a change in electron density both in the benzene and pyrrol rings. It should be noted that, whereas the red shift caused by the filtration is modest and comparable to the dilution effects, that caused by a direct interaction with TiO_2_ is much larger. The reduction in intensity for the filtrated BSA is simply a dilution effect, since the protein concentration is ~3.3-fold lower in that sample than in the BSA (control) and BSA+TiO_2_ samples. However, it is striking that the fluorescence intensity of the BSA+TiO_2_ sample is almost seven-fold smaller than that of BSA in water at the same concentration (Figure 3). This loss of intensity can be explained by fluorescence quenching, since it has also been reported that, apart from causing a red shift, TiO_2_ nanoparticles can quench fluorescence upon the formation of a higher-order complex [54]. It can, therefore, be assumed that the observed large red shift and loss of intensity in the spectrum of the BSA+TiO_2_ sample is caused by the TiO_2_ molecules reaching both the pyrrol and benzene rings of the Trp residues. In contrast to the BSA+TiO_2_ sample, the relatively short interaction of the BSA with TiO_2_ during filtration results in no red shift and no fluorescence quenching. This means that the TiO_2_–BSA interaction during filtration is either too short to be effective on the Trp residues, and/or the effect is reversible. The relatively small difference between the emission maxima of the BSA and filtrated sample can be explained by a small amount of TiO_2_ getting in the waste during filtration.

### 2.5. EPR Spectroscopy

The shape of the continuous-wave (CW) EPR spectrum of the spin labels attached to biological molecules is directly sensitive to the speed, amplitude, and symmetry of the rotational dynamics of the label constrained by the orienting potential of its environment [55,56,57,58]. We have utilised both known types of labelling targets in BSA: fatty acid bound to the protein [27] and Cys residues with a free sulfhydryl group [21,37]. We have used 5-SASL as a spin-labelled fatty acid analogue, and 5-MSL and MTSL as covalent labels of the Cys residues. These spin labels are shown in Figure 4. 

All three labels have some solubility in water, and, since unbound labels were not removed from the EPR samples, we recorded their spectra also in the absence of BSA and TiO_2_ in order to identify the different spectral components in the protein-labelled samples. It should be noted that, since TiO_2_ is diamagnetic, it did not contribute at all to the EPR spectra of the spin labels, as expected.

#### 2.5.1. Spin Labelling with 5-SASL

Figure 5 shows the CW EPR spectra of 5-SASL in water and in the three different aqueous BSA-containing samples, i.e., in the BSA, BSA+TiO_2_, and filtrated BSA. 

The spectra are normalised to the same integrated intensity (second integral), so they represent the same number of spins. The 5-SASL in water has a single component EPR spectrum of three sharp lines, as expected for a spin label freely rotating in a solvent (see, e.g., [30]). The protein-containing samples are qualitatively similar to those in previous EPR reports on spin-labelled fatty acids in the presence of serum albumins [20,27,28,30,33], showing composite spectra with different contributions from a mobile and an immobile component. Since the mobile component was identical with that of 5-SASL in water, optimised subtraction allowed us to determine the shape and relative contribution of the spectrum from 5-SASL bound to BSA, and also the mobile component (see [59] for a detailed description of the technique). The spectra of BSA and BSA+TiO_2_ have almost only the immobile component, with only a few percentages of the mobile component. In the spectrum of the filtrated BSA, the mobile component is apparently the dominant one (providing ~2/3 of the integrated intensity). However, it should be kept in mind that the BSA concentration in that sample is 3.3-fold lower than in the other two BSA-containing sample types. The relative contributions of the mobile and immobile components are given in Table 4.

Considering the lower protein concentration in the filtrated BSA sample, we can conclude that all BSA-containing samples bind the same (high) amount of 5-SASL per protein. With the exception of the filtrated BSA sample, the 5-SASL spectra are dominated by either the mobile or the immobile component to an extent that the minor component is too weak for meaningful component separation. The mobile and immobile spectral shapes require different spectrum analyses in order to obtain data on the rotational dynamics of the doxyl group of 5-SASL: The sharp hyperfine lines of the mobile component (the dominating spectra of the BSA in water and filtrated BSA) can be used to derive the mean rotation correlation time using the Kivelson formula (Equation (2)) [60,61].
(2)$$\tau_{R}\left(ns\right)=0.65\cdot W_{0}\left(Gauss\right)\cdot \left(\sqrt{\frac{h_{0}}{h_{-1}}}-\sqrt{\frac{h_{0}}{h_{+1}}}\right)$$
where τ_R_ is the rotation correlation time in ns, W_0_ is the line width of the narrowest (central) peak, and h_+1_, h_0_, and h_−1_ are the intensities of the hyperfine lines. On the other hand, if the rotational dynamics are slow (on the EPR time window) or limited in amplitude, then the EPR spectrum shows an anisotropic spread between the line positions at the minimum and maximum hyperfine-splitting values, corresponding to the z-axis of the doxyl group being oriented perpendicular or parallel, respectively, to the magnetic field. In our experiments, the immobile components represent rotational dynamics constrained by the local environment of the spin label. In this case, the orientational order determines the inner and outer hyperfine splitting (A_zz_), from which the orientational order parameter can usually be determined (see, e.g., [61,62]). However, the immobile components do not sufficiently expose the inner splittings and only the outer splitting constant can be easily determined (which is half of the magnetic field difference between the first local maximum and last local minimum); they are still in a monotonic relationship with the order parameter, with a larger outer splitting meaning a higher order. The rotation correlation times and outer hyperfine-splitting constants are also reported in Table 4. The rotation is very fast for water and for the filtrated BSA, and the values (0.12062 ns and 0.13124 ns, respectively) are very close to each other, meaning that the mobile component in the filtrated BSA spectrum corresponds to the unbound free 5-SASL. The presence of a strong immobile component in the BSA-containing samples suggests that the fatty-acid-binding sites are preserved both in the presence of TiO_2_ and after filtration. Similar outer-splitting values (64.6 G and 65.1 G) were obtained for BSA and BSA+TiO_2_. There is some uncertainty with regard to the outer splitting for the filtrated BSA because of the big signal/noise ratio of the decomposed spectrum, but it is safe to say it is ~2 G smaller for this sample. This means more disorder for the 5-SASL bound to BSA in the presence of TiO_2_, which may be an indication to loosen the fatty-acid-binding pocket of the BSA by TiO_2_. 

#### 2.5.2. Spin Labelling with 5-MSL

BSA contains 35 Cystein residues. However, according to the experimental structure of the protein (illustrated in Figure 1) and the literature data, only one (Cys34) is not involved in forming S-S bridges, with a non-bonded sulfhydryl group offering a single unique site for covalent labelling with maleimide-type spin labels [28,63,64,65,66]. The EPR spectra of 5-MSL added to water and the different BSA samples are shown in Figure 6A. 

According to its chemical structure (Figure 4B), the covalent binding of 5-MSL to a Cystein yields a relatively rigid connection between the N-O bound (bearing the unpaired electron) and the protein backbone; hence, an immobile component is expected for a covalently bound 5-MSL [21,22,28,34,67]. The EPR spectra in Figure 6A have very similar shapes but different intensity. (It should be kept in mind that the spectra represent the same number of spins; therefore, smaller amplitudes indicate line broadening.) However, there is a clear sign of a weak immobile component present in the spectra of the BSA and BSA+TiO_2_ samples (and, to lesser extent, in the filtrated BSA, which, however, has a 3.3-fold lower protein concentration), as indicated by the arrows and in the bottom part of the figure. We could separate the mobile and immobile components and analyse them in a similar way as with the 5-SASL spectra, and the extracted parameters are presented in Table 5.

Comparing the corresponding immobile fractions for 5-SASL (Table 4) and 5-MSL (Table 5), it is evident that the labelling of BSA is more efficient with 5-SASL than with 5-MSL. The immobile fraction, the rotational correlation time, and the outer splitting values are very close to those of the BSA and BSA+TiO_2_ samples, whereas the filtrated BSA has an apparently much smaller immobile component (but with a smaller outer splitting, similarly to 5-SASL). However, if we correct for the 3.3-fold lower protein concentration in the filtrated BSA than in the other two BSA-containing samples, the immobile component per protein is the same. It should be noted that the rotational correlation times are at the fast limit of the EPR time window for spin labels [55,68]. These values most probably correspond to the free rotation of the spin label in water (5-MSL is much smaller than 5- SASL). Again, the reduced fraction of the immobile component in the filtrated sample is mostly caused by the similarly lower concentration of BSA in the filtrated sample than in the BSA and BSA+TiO_2_ samples. The variation in τ_R_ is likely to be caused by the colliding of the unbound label with the surface of the slow-tumbling BSA (reducing the mean τ_R_), which agrees with the observation that τ_R_ is smaller in the filtrated BSA sample, in which the BSA concentration is smaller than in the other two BSA-containing samples. A similar explanation applies when comparing the 5-MSL in water vs. in the BSA samples. It should be also noted that the outer splitting (hence, the orientational disorder) is comparable to those obtained with 5-SASL.

#### 2.5.3. Spin Labelling with MTSL

Spin-labelling with MTSL was performed in the same way as with 5-MSL. MTSL (Figure 4C) has a disulfide bridge between the doxyl ring and the free sulfhydryl group of the Cysteine amino acid, which provides more rotational freedom and flexibility to this label compared with 5-MSL. It is, therefore, expected that MTSL displays higher mobility than 5-MSL. The EPR spectra of MTSL in water and the BSA-containing samples are shown in Figure 7. 

Through optimised spectral subtractions, we found that the MTSL spectra have three components: in addition to the expected mobile and an immobile component, we observed a five-peak component known to originate from the dimeric (biradical) form of this type of label [69]. Fortunately, the five peaks of the dimeric form do not overlap with the three lines of the immobile component of the normal, monomeric form. Therefore, by using diluted samples to change the contribution of the five-peak component (not shown), the three components could be separated: We first obtained the pure five-peak component from the series of spectra of MTLS at different label concentrations. Then, the five-peak component was subtracted from the composite spectra, leaving the normal mobile plus immobile components, which were then treated with the routine technique for the two components [59]. Although we see the immobile component as observed earlier [31], it is almost undetectable; hence, the outer splittings could not be determined. (Again, it should be kept in mind that the filtrated BSA sample has a 3.3-fold lower protein concentration than the other two BSA-containing samples.) Table 6 reports the fractional contribution of the mobile and immobile component and the rotational correlation time derived from the mobile component of the monomeric MTSL in water and in the BSA-containing samples.

According to its mobile component, the rotation of the monomeric MTSL is ~5–6 times slower in the BSA-containing samples than in water, and ~5–8 times slower than the 5-MSL in the corresponding samples. We cannot exclude the idea that the mobile components are from bound MTSL, but it is more likely that the mobile component is from MTSL monomers in water, and the effective rotational correlation time is increased in BSA-containing samples by diffusional collisions with the protein micelles, which is confirmed by the apparent concentration dependence of τ_R_ in the dilution experiment (not shown). As opposed to the results with 5-MSL (where the BSA+TiO_2_ and BSA spectra report similar rotational dynamics, as shown in Table 5), the MTSL spectrum of the BSA+TiO_2_ yields somewhat slower rotational correlation time than those of BSA or filtrated BSA (Table 6). This is probably due to the higher accessibility of TiO_2_ to bound MTSL because of its longer linker, or to its preferable interaction in water.

## 3. Materials and Methods

### 3.1. Chemicals and Sample Preparation

Bovine serum albumin (BSA) and TiO_2_ Aeroxide P25 were of analytical grade and purchased from VWR International (Debrecen, Hungary). Pristine and modified Polyvinylidene fluoride (PVDF) with molecular weight cut-off (MWCO) values of 100 kDa and ultrafilter (UF) membranes Kynar 400 PVDF with 100 kDa molecular weight cut-off (MWCO) were purchased from New Logic Research Inc. (Minden, NV, USA). The spin labels, 5-doxyl stearic acid (5-SASL) and 3-maleimido-proxyl (5-MSL), were purchased from Sigma (Budapest, Hungary), and 1-(oxyl-2,2,5,5-tetramethylpyrrolidine-3-methyl) methanethiosulfonate (MTSL) was obtained from Santa Cruz Biotechnology (Dallas, TX, USA). Ethanol was obtained from Molar Chemicals (Halásztelek, Hungary).

The aim of ultrafiltration experiments was to investigate the effect of TiO_2_ coating on filtration performance of composite membranes. The ultrafilter membranes were prepared by coating PVDF with inorganic TiO_2_ nanoparticles. Commercial PVDF membranes were used to prepare nanoparticle-coated membranes by a physical deposition method [4]. For this purpose, 0.04 g of commercial TiO_2_ was added to 100 mL of ultrapure water and ultrasonicated for 3 min. Then, the ultrasonicated suspension was filtrated through a membrane in a dead-end filtration device (Millipore, XFUF04701, Merck KGa, Darmstadt, Germany) at 0.3 MPa and dried for 1 h at room temperature before use. The procedure of the filtration has been performed as described previously [4]. Briefly, BSA rejection tests were performed at 0.1 MPa. In each filtration, 250 mL of water or model solution was filtrated until a 200 mL of permeate was obtained. Concentration of BSA was measured before and after filtration by measuring chemical oxygen demand (COD) of the solution, and BSA rejection was calculated. To check the potential effect of TiO_2_ nanoparticles on BSA filtration performance, another sample was prepared as follows: 0.04 g TiO_2_ and 250 mg BSA were dissolved in 250 mL ultrapure water and stirred (350 rpm) for 1.5 h, then filtrated through a 0.22 µm cellulose acetate (CA) syringe filter to separate BSA from TiO_2_. The clear BSA solution was then filtrated through the unmodified PVDF ultrafilter membrane (labelled as BSA/TiO_2_@PVDF), and COD rejection was calculated. Rejection was calculated using the following equation:(3)$$R\left(\%\right)=\frac{c_{0}-c}{c_{0}}\times 100$$
where c_0_ and c are the concentrations of feed and permeate solutions, respectively.

In order to examine the potential effect of TiO_2_ on BSA, four types of samples were analysed with EPR and fluorescent spectroscopy: (1) BSA 6 × 10^−5^ M in water, (2) TiO_2_ 2 × 10^−2^ M in water, (3) BSA-TiO_2_ mixture in water with [BSA] = 6 × 10^−5^ M and [TiO_2_] = 2 × 10^−2^ M in water, and (4) BSA 6 × 10^−5^ M filtrated through composite ultrafilter membranes. The bandgap of TiO_2_ P25 is around 3.2 eV, which means that 385 nm UV light can activate it, but it has poor activity under visible light [70]. The effect of light exposure was checked and excluded during experiments; nevertheless, the filtration experiments were performed in dark conditions.

The spin labels were dissolved in ethanol (6 × 10^−3^ M) for the stock solutions. Protein concentration was checked by Lowry method [71]. The BSA concentration was always reduced by the filtration; concentrations for filtrated BSA stocks were determined to be typically ~1.8 × 10^−5^ M. However, the protein concentration in the (unfiltrated) BSA and BSA+TiO_2_ samples were not adjusted to that of filtrated ones because we wanted to avoid any change in the original samples.

### 3.2. Experimental Procedures and Data Analysis

COD was measured by the standard potassium dichromate oxidation method using standard test tubes (Lovibond, Tintometer Gmbh, Dortmund, Germany) and digestion and COD measurements were carried out in a COD digester (Lovibond, ET 108, Tintometer Gmbh, Dortmund, Germany) and a COD photometer (Lovibond PC-CheckIt, Tintometer Gmbh, Dortmund, Germany).

Dynamic light scattering measurements for BSA were made by a Zetasizer Nano ZS (Malvern Panalytical, Malvern, UK) instrument. The principle of the DLS technique is that fine particles and molecules that are in Brownian motion diffuse at a speed relative to their size. To measure the diffusion speed, the scattered light of a He-Ne laser (633 nm) illuminating the particles is recorded. The light intensity at a specific angle fluctuates with time, which is recorded using a sensitive avalanche photodiode detector. The autocorrelation function of this time-series curve is informative in terms of the size distribution of the hydrodynamic radius of the particles [72]. The samples contained 0.03, 0.3, 0.6, and 6 × 10^−5^ M BSA in water.

Fluorescence measurements were carried out by a Fluorolog-3 (FL3-222) modular spectrofluorimeter (Horiba (Jobin Yvon), Kyoto, Japan) with an excitation wavelength of 295 nm. Stocks and 30-times-diluted solutions of BSA, BSA+TiO_2_, and filtrated BSA were used for the measurements.

Continuous-wave EPR measurements were carried out on a Bruker ELEXSYS-II E580 X-band spectrometer at room temperature, with the following instrument settings: scan range 3300–3400 G; microwave frequency ~9.42 GHz; microwave power 9.464 mW; modulation frequency 100 kHz; modulation amplitude 0.4 G; resolution of field axes 1024; to scans 16; sweep time 40.96 s. The first series of experiments had the molar ratio of protein and spin labels 1:2 where the concentration of the spin label was 1.2 × 10^−4^ M with the volume ratio of 2%. Further sample series were 5- or 10-fold-diluted either for both the stocks and the spin labels, hence preserving all the molar ratios, or only for spin labels. There was no column chromatography or other separation technique used to remove unbound spin labels to avoid any further agitation of the samples.

Data analysis and fitting the plotting were performed with *Igor Pro*, version 8.02 (Wave Metrics, Lake Oswego, OR, USA). Molecular graphics were made with *YASARA,* version 21.12.19 [73].

## 4. Conclusions

Our previous results on the effect of coating filter membranes with TiO_2_ nanoparticles on the membrane and filtration [4,5] could not explain the changes in the BSA structure observed here. Although filtering causes a 3.3-fold reduction in the BSA concentration, it is still significantly above the CMC. Therefore, the observed effects, as reported by EPR and fluorescence spectroscopy, are not caused by changes in the particle sizes. We can also exclude an indirect structural effect of TiO_2_ on BSA through the pH, because the observed pH changes in the BSA+TiO_2_ and filtrated BSA relative to the BSA samples are too small (<0.4 units) to be effective [43]. Unfolded protein monomers represent a negligible phase of BSA as opposed to the large micelles or assemblies present in all the samples over the wide range of size distribution. These large particles represent low mobility in the EPR time scale, which is reflected by the immobile component of the EPR spectra of the tightly bound fraction of the spin labels (5-SASL and 5-MSL). The potentially more loosely bound MTSL has a negligible bound fraction. It is important to note that the source of the fluorescence signal and the EPR of spin-labelled fatty acids are not as specific as the EPR of the uniquely unblocked Cys34, since there are two Trp residues and up to seven fatty-acid-binding pockets in BSA. Indeed, there are indications that all the binding sites in an albumin may have bound fatty acid even at the lowest levels of ligand loading [31]. In addition, Cys34 (in domain I) is relatively close to the protein surface [65,66], and, when spin-labelled, it has been reported to be more accessible to collisional interaction with a water-soluble paramagnetic relaxant than the acyl chain of non-covalently bound spin-labelled fatty acids (located in hydrophobic channels in the protein) [28]. Here, we found that the Trp regions are more sensitive to the presence of TiO_2_ nanoparticles, whereas both the stearic-acid-binding pockets and the environment of the spin-labelled Cysteine are more sensitive to the mechanical stress on BSA caused by the filtration process: the labels are more mobile in these samples, despite a comparatively very similar level of binding. The difference between these processes (as compared to the BSA dissolved in water) is that filtration exerts a mechanical stress on the protein and its aggregated forms (micelles) with a relatively short duration of contact with the TiO_2_. On the contrary, mixing BSA with TiO_2_ does not result in a mechanical stress to the BSA, but the protein is permanently exposed to interaction with TiO_2_. The visual appearance of the BSA solutions suggests that there is no precipitation of BSA. In addition, our spectroscopic data show that BSA has a native-like fluorescence after filtration and retains the control level of the binding of fatty acid and 5-MSL spin label both after filtration and in the presence of TiO_2_, evidencing that the structural integrity of the BSA is preserved in the present experiments.

It can be concluded that the filtration process and prolonged exposure to TiO_2_ have a significant effect on different regions of BSA. However, the unfolding, denaturing, or precipitation of the protein by the prolonged presence of TiO_2_ or filtration through a TiO_2_-coated filter membrane was not observed. Obviously, further research is needed, for instance, in the direction of slower filtering and/or different coating materials with higher elimination efficiency, filtration, and TiO_2_ effect on the particle size of BSA and its binding of fatty acids and CD spectroscopy of changes in the secondary structure.

## Figures and Tables

**Figure 1 molecules-28-06750-f001:**
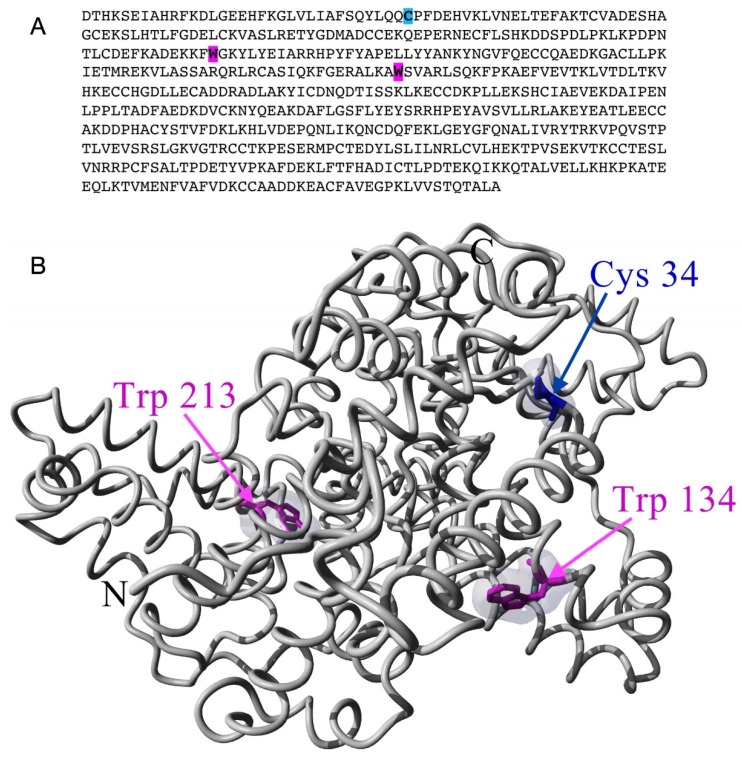
The amino acid sequence (**A**) and an illustrative 3-dimensional structure (**B**) of BSA (PDB i.d. 4s5f). The autofluorescence residues (W, Trp) and the spin-label binding site (C, Cys) are highlighted and coloured in magenta and blue, respectively.

**Figure 2 molecules-28-06750-f002:**
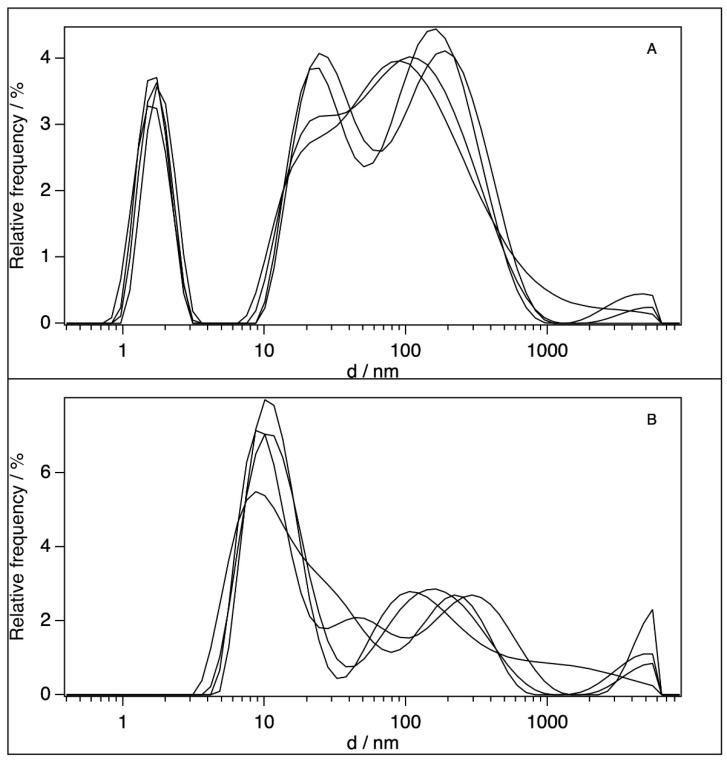
Particle size distribution in BSA solutions in water: [BSA] = 6 × 10^−5^ M (**A**), and [BSA] = 6 × 10^−7^ M (**B**). The different curves represent independent experiments.

**Figure 3 molecules-28-06750-f003:**
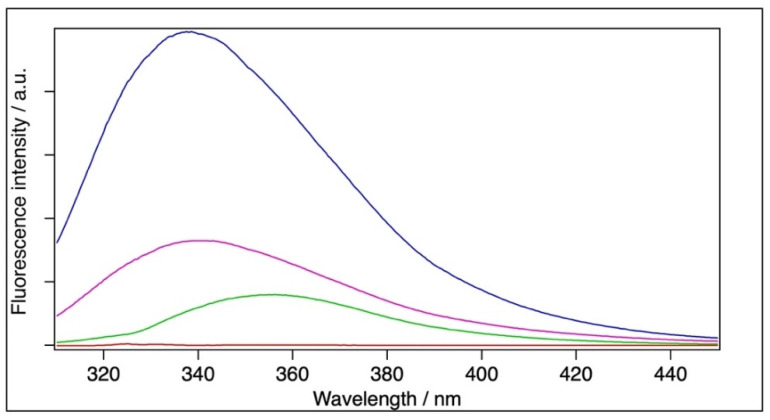
Representative fluorescence spectra. Blue: BSA ([BSA] = 2 × 10^−6^ M), magenta: filtrated BSA ([BSA] = 6 × 10^−7^ M), green: BSA+TiO_2_ ([BSA] = 2 × 10^−6^ M, [TiO_2_] = 6.7 × 10^−4^ M), and red: TiO_2_ ([TiO_2_] = 6.7 × 10^−4^ M).

**Figure 4 molecules-28-06750-f004:**
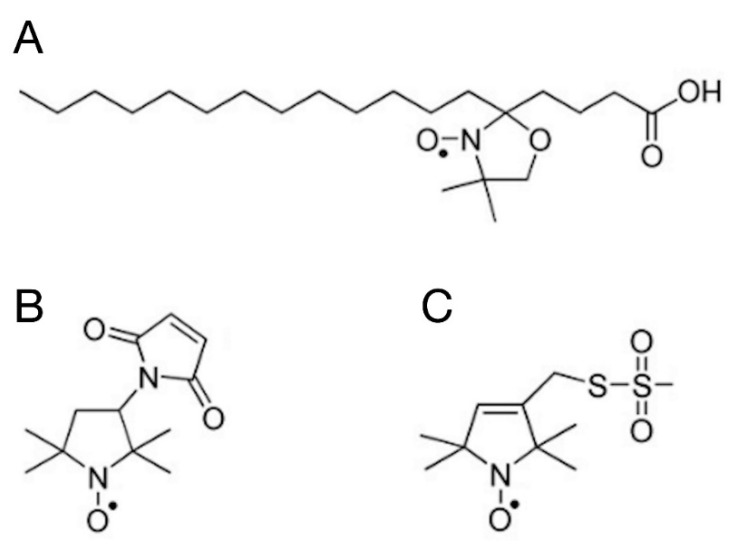
The chemical structure of the spin labels used for the EPR measurements. The EPR spectrum is originating from the unpaired electron located in a π* orbital of the N-O bound: (**A**) 5-SASL, (**B**) 5-MSL, and (**C**) MTSL.

**Figure 5 molecules-28-06750-f005:**
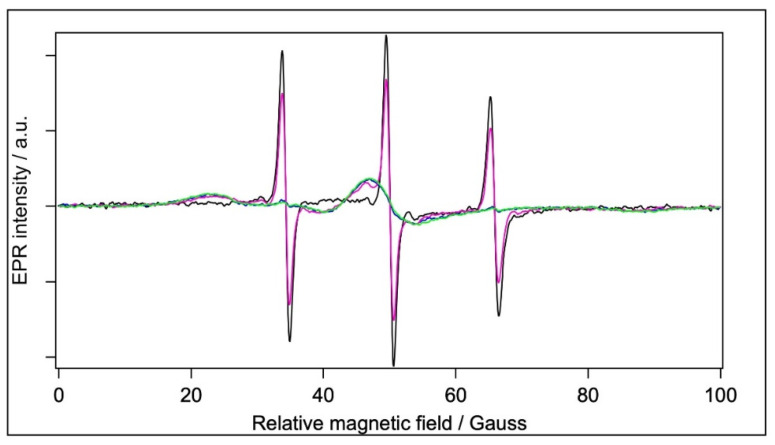
Normalised CW EPR spectra of samples labelled with 5-SASL. Black: water, blue: BSA in water, magenta: filtrated BSA, and green: BSA+TiO_2_. [5-SASL] = 1.2 × 10^−4^ M.

**Figure 6 molecules-28-06750-f006:**
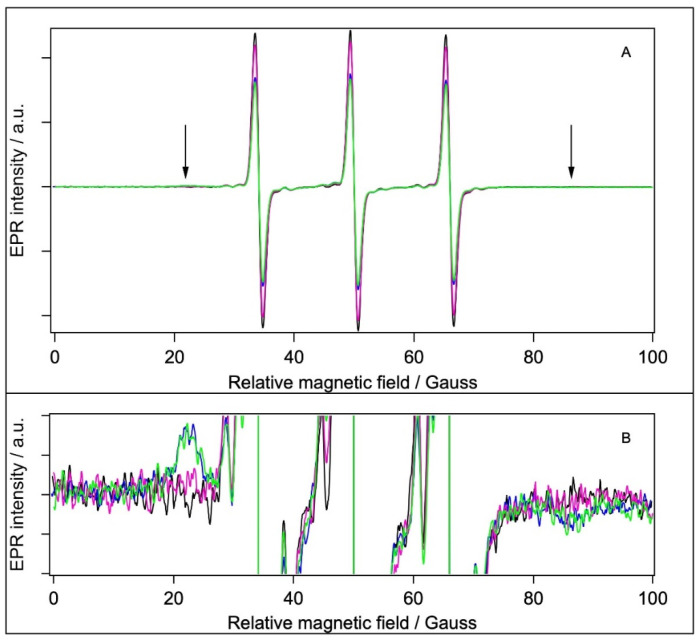
Normalised CW EPR spectra of samples labelled with 5-MSL. Black: water, blue: BSA, magenta: filtrated BSA, and green: BSA+TiO_2_. (**A**) Full spectra; (**B**) 73-fold-magnified spectrum to visualise the immobile component (shown with arrows on panel A). [5-MSL] = 1.2 × 10^−4^ M.

**Figure 7 molecules-28-06750-f007:**
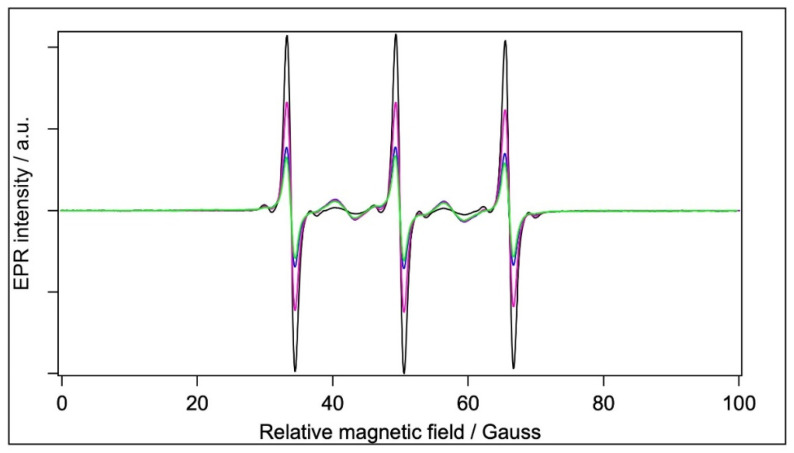
Normalised CW EPR spectra of samples labelled with MTSL. Black: water, blue: BSA, magenta: filtrated BSA, and green: BSA+TiO_2_, [MTSL] = 1.2 × 10^−4^ M.

**Table 1 molecules-28-06750-t001:** BSA retention during filtration of BSA through PVDF or PVDF/TiO_2_ composite membrane.

Sample	[BSA]/M	Membrane	R%
BSA	1.5 × 10^−5^	PVDF	92
BSA	1.5 × 10^−5^	PVDF/TiO_2_	28
BSA(TiO_2_)	1.5 × 10^−5^	PVDF	91
BSA(TiO_2_)	4.9 × 10^−6^	PVDF	37
BSA	4.9 × 10^−6^	PVDF	37

**Table 2 molecules-28-06750-t002:** Particle size of BSA.

[BSA]/10^−6^ M	d1/nm	d2/nm	Particle Type	Medium	Reference
0.6		11	dimers (d2)	H_2_O	own result
0.65		8.9	compact aggregates (d2)	pH 7.2, 10 mM phosphate buffer	[16]
4.5	10		monomer (d1)	pH 7.0, H_2_O	[17]
5	7.3	13.5	monomer (d1), dimer (d2)	pH 7.4, H_2_O	[46]
10	3.4		monomer (d1)	pH 7.0, H_2_O	[18]
25		10.6	compact aggregates (d2)	pH 7.2, 10 mM phosphate buffer	[16]
60	1.7	24–200	unfolded monomers (d1) large aggregates (d2)	H_2_O	own result
100	5.2		undefined particle (d1)	pH 7.4, H_2_O	[47]
120.4	2	22	denaturated monomers (d1), aggregation of denaturated monomers (d2)	H_2_O	[48]
120.4		12.4	dimer (d2)	pH 7.4, 0.01 M PBS buffer	[48]

**Table 3 molecules-28-06750-t003:** Tryptophan emission maxima.

Sample	[BSA]/M	Fitted Emission Maximum (λ_max_)/nm	Mean Emission Maximum (<λ_F_>)/nm
BSA	6 × 10^−5^	338.9	352.3
Filtrated BSA	1.8 × 10^−5^	340.2	356.9
BSA+TiO_2_	6 × 10^−5^	-	-
BSA	2 × 10^−6^	338.4	352.3
Filtrated BSA	6 × 10^−7^	340.7	355.0
BSA+TiO_2_	2 × 10^−6^	355.6	363.3

**Table 4 molecules-28-06750-t004:** EPR parameters of samples spin-labelled with 5-SASL. [5-SASL] = 1.2 × 10^−4^ M.

Sample	Mobile Component/%	Immobile Component/%	τ/ns	2A_zz_/G
Water	100	-	0.1206	-
BSA	2.4	97.6	-	64.62
Filtrated BSA	71	29	0.1312	62.3–64.1
BSA+TiO_2_	1.0	99.0	-	65.05

**Table 5 molecules-28-06750-t005:** EPR parameters of samples spin-labelled with 5-MSL. [5-MSL] = 1.2 × 10^−4^ M.

Sample	Mobile Component/%	Immobile Component/%	τ/ns	2A_zz_/G
Water	100	-	0.0043	-
BSA	71	29	0.0082	63.92
Filtrated BSA	93	7	0.0056	cc. 61 G *
BSA+TiO_2_	67	33	0.0084	64.20

* The immobile component of this sample is very noisy (because of its small contribution to the composite spectrum); hence, the outer splitting is much less certain than for the other samples.

**Table 6 molecules-28-06750-t006:** EPR parameters of samples spin-labelled with MTSL. [MTSL] = 1.2 × 10^−4^ M.

Sample	Mobile Component/%	Immobile Component/%	τ/ns	2A_zz_/G
Water	100	-	0.0094	-
BSA	97–98	2–3	0.0303	-
Filtrated BSA	100	-	0.0224	-
BSA+TiO_2_	97–98	2–3	0.0536	-

## Data Availability

All experimental data presented in this study are available upon request from the corresponding author.

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
