# Peer review of "Spin-Label Electron Paramagnetic Resonance Spectroscopy Reveals Effects of Wastewater Filter Membrane Coated with Titanium Dioxide Nanoparticles on Bovine Serum Albumin"

_molecules, 2023, doi:10.3390/molecules28196750_

Round 1

Reviewer 1 Report

The ms describes experiments designed to determine the effect of TiO2 nanoparticles and filtration on BSA. The abstract states that the filtering process and prolonged exposure to TiO2 nanoparticles had significant effects on different regions of BSA. This reviewer disagrees that filtering affects BSA itself, it looks like filtering just separates BSA monomers and oligomers. It is not clear what is the significance of the observed effect of TiO2 on BSA intrinsic fluorescence.

Table 1 sets up the goal of the ms. Two concentrations of BSA are used, treated and not treated with TiO2 nanoparticles. The membrane was also treated with nanoparticles. According to Table 1 BSA treatment with nanoparticles does not change BSA retention upon filtration, but treatment of the membrane does. Therefore, the question appears, why study BSA and not the membrane? Was the membrane integrity checked after the treatment? It looks like the membrane becomes leaky after the treatment. Different retention of BSA clearly reflects protein oligomerization at increased concentration since the mw of BSA monomer is 66.5 kDa and the membrane’s MWCO is 100kDa. It is not clear why the not very accurate COD method was used to determine BSA concentration to calculate retention, and not optical absorption or Lowry or Bradford method.

Figure 2 shows particle size distribution in BSA solutions of low and high concentrations. At high concentration particle size 1.7nm was determined. BSA monomer has diameter approx.. 7nm (4f5s.pdb) and this reviewer wonders how micelles can be smaller than monomeric protein.

Observed BSA+TiO2 fluorescence red shift means some sort of effect on BSA fluorescence, it is not clear if this effect is relevant to BSA function and if this effect is irreversible upon TiO2 removal.

EPR, SASL – apparently BSA filtration affects BSA-SASL binding, more info needed on the strength of BSA-SASl interaction. TiO2 does not affect BSA-SASL interaction.

EPR, 5-MSL data – assuming that free label is removed after labeling (it is not clear from the Materials and Methods section) the result can be explained as a separation of monomers and oligomers of labeled BSA. Apparently oligomers produce some restrictions to spin label motion and this is why immobile component eliminated in the flow through. It appears that TiO2 does not have any effect on BSA spin label-wise. Small differences of mobile and immobile fractions are small and authors need to make a statistical analysis to draw any conclusions (see Table 5).

Figure 7 shows the effect of TiO2 on BSA, since just BSA in solution and BSA-TiO2 mix produce different spectra. The difference is not discussed in the ms. It is not clear if this effect is relevant to BSA function or irreversible upon TiO2 removal.

The ms struggles from typos (prolongued, 4s5f, TiO1, noticable), awkward phrases (“The fluorescent residues in BSA offer a label-free technique via autofluorescence”) and jargon (the "other" component).

Author Response

We thank very much for the time and effort of the reviewer for reading our manuscript thoroughly and providing important comments to improve it, also providing great ideas for further experiments in the future. Below are our point-by-point responses to the reviewer's comments (all changes are marked in yellow in the revised manuscript):

•1• "... it looks like filtering just separates BSA monomers and oligomers"

-> Answer: We come back to this point under •9• below.

•2• "... It is not clear what is the significance of the observed effect of TiO2 on BSA intrinsic fluorescence"

-> Answer: The fluorescence data prove that nanoparticles reach Trp residues because of the observed red shift and the strong quenching, on the one hand. The lack of strong red shift and any quenching in the filtered sample (BSA after contacting the TiO2-coated membrane) suggests that the TiO2 effect on the Trp residues is probably reversible, on the other hand. To clarify this point more, we have added several sentences (citing an important new reference https://doi.org/10.1016/j.colsurfa.2021.127253 ) under Fig. 3, starting as "It is striking that the fluorescence intensity of the BSA+TiO2 sample is almost 7-fold smaller than that of BSA in water at the same concentration ...". (Because of two new references, the list of references are re-numbered.)

•3• "... why study BSA and not the membrane?"

-> Answer: We have previously studied the filter membranes as well as the mechanism of BSA fouling. These results were published earlier (see Jigar et al., 2020; Sisay et al., 2023, already cited in the manuscript). In the present series of experiments the aim was to check if there was any change in the protein structure due to filtration or contact with the nanoparticles. This is mentioned in the Introduction and we have added a new sentence to the beginning of the Conclusions to make this more clear, starting as "Our previous results about the effect of coating the filter membranes with TiO2 nanoparticles on the membrane and the filtration...".

•4• "Was the membrane integrity checked after the treatment?"

-> Answer: In our above mentioned related experiments the surface characteristics of the treated membranes (AFM, SEM) were examined before and after filtration. Those results and the stability of flux after photocatalytic cleaning of the covered membranes show that the membrane structure remains intact. We now clearly state this in the first paragraph of the Introduction.

•5• "It is not clear why the not very accurate COD method was used to determine BSA concentration to calculate retention, and not optical absorption or Lowry or Bradford method."

-> Answer: The aim of the development of these types of membranes is to use them for dairy wastewater treatment. In case of real or even model wastewater the COD is a routinely and widely used relevant parameter to characterise the amount of pollutants (not only for proteins). Thus, this method was used for determination of filtration efficiencies, also in order to be compatible with other studies. Nevertheless, we did use the Lowry assay to determine protein concentration before and after filtering and it is stated in the manuscript.

•6• "Figure 2 ... particle size 1.7nm was determined. BSA monomer has diameter approx.. 7nm (4f5s.pdb) and this reviewer wonders how micelles can be smaller than monomeric protein"

-> Answer: We thank the reviewer for pointing out this discrepancy (actually a mistake on our part)! When preparing the manuscript have evaluated the literature about DLS of BSA in aqueous environment and came to the conclusion that (i) the particle size distribution strongly depends on the BSA concentration and (ii) the reports are very diverse regarding particle sizes and their assignments (cf. Table 2). For instance, surprisingly several studies assign the < 4 nm peaks to some forms of monomeric BSA. A full and detailed study would certainly be timely and relevant about how the particle distribution of BSA changes under different condition (we mention this now in the end of the Conclusions and we will propose it to our collaborating partner who owns the DLS setup). Since it would have been beyond the scope and intention of this manuscript, we have limited ourselves to the test particle size distribution below and above the cmc. This was rational because it was obvious that the filtration was to lead to decreased BSA concentration. It turned out that BSA was above the cmc both pre- and post filtration, hence Fig. 2A applies to both situations. (Fig. 2B presents data below the cmc, where dimers clearly dominate – we have now corrected our assignment in Table 2.). That means we can exclude changes in the particle sizes distribution induced merely by concentration effects, and that was our purpose with the DLS experiments. Regarding the ~1.7nm peak, we found one reference (Burgstaller et al., 2018, cited in the manuscript) reporting quite the same distribution profile as our (Fig. 2A), except that in our case the larger aggregates reach much higher diameters. They speculated that the < 3 nm peak is the result of unfolded monomers, which behave differently in DLS experiments than globular protein particles. Therefore, it was a mistake on our part to mark that peak as "micelles", and we have corrected it in Table 2. It should be noted that the ~1.7 nm peak in Fig. 2A has a very small, actually negligible contribution to the distribution curve (considering the logarithmic scale of the x axis). And we can say that BSA is predominantly present in multimeric assemblies or micelles both before and after filtration. We have also corrected the related text in section 2.2 of the Results (all changes are marked in yellow).

•7• "... BSA+TiO2 fluorescence red shift means some sort of effect on BSA fluorescence, it is not clear if this effect is relevant to BSA function and if this effect is irreversible upon TiO2 removal"

and

•8• "... apparently BSA filtration affects BSA-SASL binding, more info needed on the strength of BSA-SASl interaction. TiO2 does not affect BSA-SASL interaction."

-> Answer: (For the most part, we addressed this question under the 2nd point above.) Filtration (with TiO2-coated membrane) means relatively short time of BSA interacting with TiO2, This process gives no red shift and no quenching in our experiments. In contrast, BSA in the presence of TiO2 shows strongly quenched and red shifted fluorescence spectra. This means that the TiO2-BSA interaction during filtration is either too short to be effective and/or the effect (and the binding) of TiO2 is reversible. We think that reversibility is likely because the TiO2 nanoparticles are rather mobile in the filter membrane (so they could have access to Trps even during filtration similarly to when co-dispersed with BSA). In addition, reversibility is also supported by our COD experiments where we found that BSA pretreatment with TiO2 had no noticeable change on the filtration performance (after removing TiO2). We agree that more experiments would be needed to test whether TiO2 affects BSA function (binding of fatty acids), and this is now mentioned in the end of Conclusions. For instance, filtering (as a mechanical stress) might affect the fatty acid binding pockets or protein-protein interactions. However, the scale of such a conclusive study on this issue would go beyond this manuscript, and it would be a different story (the effect of TiO2 on BSA structure and fatty acid binding) not necessarily related to wastewater filtering.

•9• "EPR, 5-MSL data – assuming that free label is removed after labelling (it is not clear from the Materials and Methods section) the result can be explained as a separation of monomers and oligomers of labeled BSA. Apparently oligomers produce some restrictions to spin label motion and this is why immobile component eliminated in the flow through. It appears that TiO2 does not have any effect on BSA spin label-wise. Small differences of mobile and immobile fractions are small and authors need to make a statistical analysis to draw any conclusions (see Table 5)."

-> Answer: Although, this interpretation is very attractive and logical (we have also thought about this), we think that separation of monomers and oligomers by filtration can account for the spectral changes we see with 5-MSL because the presence of monomers is very negligible (Fig. 2A), therefore we don't think we can account any spectral component to spin-labelled monomeric BSA. In addition, free label was not removed (in order to prevent exposing the samples to concurrent filtering processes and additional un-wanted effects). Instead we considered the presence of an aqueous peak during the analysis of the  EPR spectra. We added this sentence to the end of the paragraph about the EPR method: "There was no column chromatography or other separation technique used to remove unbound spin labels in order to avoid any further agitation of the samples." We think that the sharp component is free label maybe with loose interaction with the protein (oligomers), also because it appears that the labelling efficiency was low. Please also note that filtration reduced the protein concentration 4-times, and consequently we got 4-times less immobile component.

•10• "Figure 7 shows the effect of TiO2 on BSA, since just BSA in solution and BSA-TiO2 mix produce different spectra. The difference is not discussed in the ms. It is not clear if this effect is relevant to BSA function or irreversible upon TiO2 removal."

-> Answer: We have not investigated the reversibility of TiO2 effect on BSA with the spin labels. (It seems not easy to remove TiO2 from the spin-labelled protein samples without affecting the concentration of the free label and maybe also any loose binding of it to the to the protein), that is without affecting the EPR spectrum. However, as pointed out above, we have good arguments to assume that the effect is reversible. Regarding the somewhat slower rotational dynamics in the BSA+TiO2 sample, we have added two sentences to end of section 2.4.3, starting as "As opposed to the results with 5-MSL ..." to point out the differences between 5-MLS and MTSL results. In addition, we noted a discrepancy between Fig. 7 and Table 6, and found out that we have previously not selected the most representative spectra, by mistake. (Please note that – as indicated in the text - there was a strong contribution from dimer MTSL to the spectra and the contribution of that component also varied affecting the shape of the spectrum of course. Since Fig. 7 shows the original spectra this variation of the dimer MTSL component could cause confusion). For this reason we have now updated both Fig. 7 and Table 6, without affecting the results and conclusions, and they agree now.

•11• "The ms struggles from typos (prolongued, 4s5f, TiO1, noticable), awkward phrases (“The fluorescent residues in BSA offer a label-free technique via autofluorescence”) and jargon (the "other" component)."

-> Answer: We have checked and removed typos and stylistic mistakes (including those mentioned by the reviewer) to our best.

Reviewer 2 Report

A very acute and challenging problem solvation of bio-to-surface interaction is aimed in this paper. Specifically, protein interactions and conformational behavior when filtering through TiO2 coated commercial filters is studied thoroughly. Particular attention in this study is given to utilize EPR through spin labeling together with fluorescence spectroscopy and DLS method. The paper might be of interest to the broad readership of Molecules. However, following issues should be thoroughly addressed in order to deliver more clarity, to support conclusive statements and to reveal the details for the scientific community.

1.      DLS derived diameters were discussed well in the article. However, there was no attention to zeta potentials of both TiO2 and BSA particles.

2.      All changes in protein conformations are usually perfectly registered with the use of circular dichroism method. In this regard, to support conclusion in the paper it is important to provide CD proved evidence of investigated effects.

3.      TiO2 nanoparticles are known well to absorb a UV light with following photodestruction of the organic molecules adsorbed on their surface. Was experimental work conducted in a dark conditions?

4.      Since the fluorescence intensity is in arbitrary units, it is not necessary to carry 106 multiplicator for the Y-axis on the Figure 3. Spectra can also be normalized in order to show the peak shifting.

English is fine

Author Response

We thank very much for the time and effort of the reviewer for reading our ms thoroughly and providing important comments to improve it. Below are our point-by-point responses to the reviewer's comments (all changes are marked in yellow in the revised manuscript):

•1• "... there was no attention to zeta potentials of both TiO2 and BSA particles"
-> Answer: Analysis of possible changes in Zeta-potential of the particles could be indeed relevant as it might, for instance, give further information about the direct interaction of the protein with TiO2 particles via changes in the surface charges of the particles. However, we have already demonstrated the direct interaction of TiO2 particles with the Trp residues. In addition, interpretation of changes in the Zeta-potential could be very challenging without more extensive DLS experiments on the protein and the TiO2 nanoparticles at different concentrations, and before and after interacting with the filter membrane. Whereas such experiments would certainly bring new and relevant data, it would be beyond the scope of this manuscript, in which the main focus is on local changes at certain residues using (mainly spin label EPR) spectroscopic techniques and to exclude large changes in particle size.

•2• "... provide CD proved evidence of investigated effects"
-> Answer: We fully agree that CD spectroscopy would be the most appropriate method to explore overall changes in the secondary structure of BSA before and after filtering and interaction with TiO2 nanoparticles (in we now mention it in the end of the Conclusions), although (similarly to DLS) it would require a lot more work (impossible within ten days given for revising our ms). On the other hand, we do not think that CD data could prove or dis-prove our EPR and fluorescence results and conclusions because the latter ones report about local changes, but local and global structural changes do not necessarily point in the same direction. Nevertheless, we have at least a clear hint from fluorescence that large opening (un-folding or denaturation) of BSA does not happen because in that case we should have seen much larger shifts in the emission maximum (without quenching). This is now mentioned in the manuscript, in relation to our reply to the first reviewer. In addition, this manuscript is submitted to a special issue about application of EPR spectroscopy, and adding detailed CD (and DLS) work would remove much of the focus from the demonstration of what spin label EPR technique can provide in studying such a system.

We thank the reviewer for this suggestion, and we will include CD spectroscopy in the future (in a collaboration, since we don't own this technique) on this system because it is indeed an important piece of information to what extent, if any, the secondary structure composition of BSA changes during these filtering processes or interaction with TiO2.

•3• In relation to TiO2 nanoparticles absorbing UV light, "was experimental work conducted in a dark conditions?"
-> Answer: Yes, the experiments were conducted in the dark, and we have now added the following text to the experimental section to make it clear. "The bandgap of TiO2 P25 is around 3,2 eV ... " In this text we have added a new reference. (Because of two new references, the list of references are re-numbered.)

•4• We have corrected the Figures 3 as requested, and also made the axis labels more consistent for Figures 2, 3, 5, 6 and 7.

Round 2

Reviewer 1 Report

The reviewer finds the authors' answers unsatisfactory and is additionally perplexed by certain aspects of the manuscript.

1.    Definitions: The manuscript lacks clear and concise definitions for the samples. As far as the reviewer can discern, water serves as the solvent for all the samples. Consequently, the term "BSA in water" appears to be technical jargon and requires clarification. Additionally, there is ambiguity surrounding the description of "BSA (TiO2)." Is it referring to BSA exposed to TiO2 and subsequently separated via filtration, or does it denote an actual mixture of BSA and TiO2? It is imperative that the sample names are defined with absolute clarity.

2.    It is known that the suspension of TiO2 nanoparticles typically maintains a pH level around 4.3. On the other hand, water solutions of BSA are pH-neutral. In the course of discussing the results, the authors should provide a rationale for this discrepancy in buffering conditions.

3.    The authors observed unusual penetration of BSA through filters that have been treated with TiO2 nanoparticles, and they have conducted an investigation into this phenomenon using techniques such as DLS, BSA intrinsic fluorescence, and BSA spin labeling. It is worth mentioning that in the revised ms the authors assert that membrane integrity is not a contributing factor, as indicated by their prior research. This aspect warrants a more comprehensive discussion within the manuscript. Another potential explanation for this phenomenon could be related to the structural integrity of BSA, which the authors have not explored. This reviewer recommends considering an additional analysis, such as CD spectroscopy, to assess the structural integrity of BSA in each sample.

4.    In the fluorescence study, there is ambiguity regarding the nature of the BSA-TiO2 sample. Is it BSA treated with TiO2 or a mixture with actual TiO2? If TiO2 is indeed part of the sample, the reviewer would appreciate information on the fluorescence spectrum of the TiO2 suspension. It raises the question of whether Figure 3 might depict a combination of BSA's intrinsic fluorescence and the fluorescence originating from TiO2 nanoparticles. An alternative interpretation of the observed results could involve TiO2 inducing a structural unfolding of BSA, and in this context, CD spectroscopy could be employed to provide confirmation.

5.    In spin probe experiments, MTSL's excessive mobility renders it incapable of yielding valuable insights. The outcome of SASL binding might indicate a partial unfolding of BSA, necessitating validation through CD spectroscopy. The data from the MSL probe is perplexing due to the ambiguity of sample names. It would be interesting to know how the slow and fast motional components of the spin probe spectra were separated to determine their fractions. And since free spin probe was not removed from the sample, is it any sense to discuss these components? It is not clear which EPR signal is from just a probe in solution, and which is from labeled protein. What is the labeling efficiency?

6.    Ultimately, the revisions made after the initial review have steered the manuscript in a more favorable direction. However, substantial improvements are still required for the manuscript to be considered for publication.

Author Response

We thank the reviewer for valuable new requests and suggestions to further improve our manuscript. Below are our point-by-point responses to their review:

1.    "Definitions: The manuscript lacks clear and concise definitions for the samples. ..."
-> Answer:
We have used four type of samples that were described in the Methods section, and we thought that the definitions were clear: BSA, TiO2, BSA+TiO2 (yes, BSA in the presence of TiO2) and filtrated BSA (with no extra TiO2 added). These samples made our experiments compatible with the real life filtration process and its components.
-> Changes:
Upon the request of the reviewer, we have now added a new section before the (EPR and fluorescence) spectroscopic measurements ("2.3. Samples for fluorescence and EPR spectroscopy") explaining the four type of samples and also the results of the new pH measurements (see below). In addition, we have now made the short names of the four types of samples to be fully consistent in the spectroscopy experiments (including the tables and the figures).

2.    "It is known that the suspension of TiO2 nanoparticles typically maintains a pH level around 4.3. On the other hand, water solutions of BSA are pH-neutral. In the course of discussing the results, the authors should provide a rationale for this discrepancy in buffering conditions."
-> Answer:
We did not want to use buffers to compensate for any pH changes caused by TiO2, because it is also not done when the filter membranes are used in practice for wastewater cleaning. (If TiO2 acts indirectly via pH, if at all, then it is part of the effect.) Anyway, we have now measured the pH values for all four type of samples (both at the stock and the diluted concentrations), and based on the results we can safely exclude any indirect effect of TiO2 via pH changes.
-> Changes:
The new section "2.3. Samples for fluorescence and EPR spectroscopy" of Results explains the approach, the reasoning, and presents and discusses our results on pH changes. The main message (that we can exclude pH-related effects) is also mentioned now in the Conclusions.

3.    "...It is worth mentioning that in the revised ms the authors assert that membrane integrity is not a contributing factor, as indicated by their prior research. This aspect warrants a more comprehensive discussion within the manuscript. Another potential explanation for this phenomenon could be related to the structural integrity of BSA, which the authors have not explored. This reviewer recommends considering an additional analysis, such as CD spectroscopy, to assess the structural integrity of BSA in each sample."
-> Answer:
Firstly, please note that (instead of disintegration or unfolding) we are considering different local structural changes caused by the filtration and the presence of TiO2. Secondly, even without any further study with an additional technique, we are quite certain in excluding the loss of or damage to the structural integrity of BSA due to either the presence TiO2 or the filtration process, i.e. the BSA+TiO2 and filtrated BSA samples, respectively. Briefly, according to our spectroscopic data neither filtration nor the presence of TiO2 causes unfolding or disintegration of BSA. The arguments are as follows:
(i) We have seen no sign of protein precipitation.
(ii) The red shift of Trp fluorescence emission maximum of the filtrated BSA should be much larger if either of Trp residues gets exposed to water. In other words, filtration does not cause protein unfolding under the present conditions.
(iii) Since both the 5-SALS and 5-MSL binding are retained all three type of BSA-containing samples (see below), we must assume that the quenching and the red shift in the BSA+TiO2 samples is caused by TiO2 molecules reaching at least one Trp residue, and not by protein unfolding.
(iv) It is true, that the red shift and of Trp fluorescence emission maximum and the strong fluorescence quenching of the BSA+TiO2 sample could indeed also happen if BSA unfolds. However, if we take into account the lower protein concentration in the filtrated BSA sample, BSA binds basically the same amount of fatty acid (on a per protein basis) in all three BSA-containg samples, that is: BSA, BSA+TiO2 and filtrated BSA. (The filtrated BSA binds less 5-SASL, however, in that sample the protein concentration is ~3.3-fold lower, so one BSA binds similar amount of 5-SASL as the other two samples.)
(v) Similarly, if we take into account the ~3.3-fold lower protein concentration in the filtrated BSA sample, all 3 BSA-containing samples bind 5-MSL quite similarly.
-> Changes:
We now emphasise the ~3.3-fold lower protein concentration in the filtrated BSA samples relative to the BSA and BSA+TiO2 samples in the respective parts of the Results. We also state that, as a consequence, the binding of fatty acid and the Cys labels are similar over all three BSA-contacting samples. In addition, we also address the preserved structural integrity of BSA in the Conclusions, along the above arguments.

Regarding additional CD experiments (mentioned also under points 4. and 5. by the Reviewer), we agree that CD spectroscopy would be the most appropriate method to explore overall changes in the secondary structure of BSA. Measurement would be needed at least at two concentrations, before and after filtering and interaction with TiO2 nanoparticles, and might need to exclude contribution from TiO2. (We do mention in the Conclusions CD needed in future studies.) However, such a conclusive CD study would require a lot of work and time (since we do not own this technique and would have to involve new collaborating partner and rely on the other group). We see this impossible within the provided period for responding the reviewer's comments, and such a work could even merit a new manuscript on its own.
We believe that we provide a lot of data and results from the three techniques that we have used (DLS, fluorescence, EPR) sufficient for a paper. We have now good arguments that the integrity and the main function (fatty acid binding) are not reduced significantly in the present experiments, hence CD is not needed to prove that further. In general, we do not think that CD data could prove or dis-prove our current results and conclusions because fluorescence and spin label EPR report about local changes, but local vs. global (secondary str. composition) structural changes do not necessarily point in the same direction.
Further, spin label EPR and fluorescence spectroscopy are highly appropriate methods in their own right in such protein systems, as supported by many publications in the literature (using them even as single techniques) in spectroscopic studies. This manuscript is submitted to a special issue about application of EPR spectroscopy, and adding detailed work with CD and other techniques would unfavourably shift of the focus away from demonstrating what spin label EPR technique can provide in studying such a system.

4.    "In the fluorescence study, there is ambiguity regarding the nature of the BSA-TiO2 sample. Is it BSA treated with TiO2 or a mixture with actual TiO2? If TiO2 is indeed part of the sample, the reviewer would appreciate information on the fluorescence spectrum of the TiO2 suspension. It raises the question of whether Figure 3 might depict a combination of BSA's intrinsic fluorescence and the fluorescence originating from TiO2 nanoparticles. An alternative interpretation of the observed results could involve TiO2 inducing a structural unfolding of BSA, and in this context, CD spectroscopy could be employed to provide confirmation."
-> Answer:
Yes, the BSA+TiO2 sample means BSA in the presence of TiO2 (mixed in water). We have measured the fluorescence of TiO2 (under the very same conditions as the BSA samples) and TiO2 has invisible fluorescence contribution to the Trp spectra of from BSA. In addition, the fluorescence spectrum of the BSA+TiO2 sample is basically identical with that in ref. (Pandit et al., 2021) identified as direct TiO2 effect on Trp of BSA. We can exclude structural unfolding of BSA by TiO2 (under the present conditions), see above.
Regarding CD, see above too.
-> Changes:
- We have made the sample definitions very clear in a new section "2.3. Samples for fluorescence and EPR spectroscopy" before the fluorescence and EPR results (and they are still also mentioned in the Methods section).
- Because of the new section, the subsequent sections are re-numbered.
- We now use the names and short names of the samples very consistently throughout the manuscript.
- We have added the fluorescence "spectrum" of TiO2 to Fig. 3, and mention in the text that there is no contribution from TiO2 to the Trp fluorescence from BSA. (Similarly, we also state the TiO2 does not have any EPR contribution to the spin label's spectra.)

5.    "In spin probe experiments, MTSL's excessive mobility renders it incapable of yielding valuable insights. The outcome of SASL binding might indicate a partial unfolding of BSA, necessitating validation through CD spectroscopy. The data from the MSL probe is perplexing due to the ambiguity of sample names. It would be interesting to know how the slow and fast motional components of the spin probe spectra were separated to determine their fractions. And since free spin probe was not removed from the sample, is it any sense to discuss these components? It is not clear which EPR signal is from just a probe in solution, and which is from labeled protein. What is the labeling efficiency?"
-> Answer:
- Yes, MTSL is very mobile. The immobile component is certainly bound but it is so small that it renders its further analysis impossible. The dominant mobile component is either loosely, non-covalently bound (or just adsorbed) to the surface of BSA or it is free MTSL. It would need too much material, time and effort to determine which is the case. Nevertheless, this component does show difference for BSA vs BSA+TiO2 vs filtrated BSA. This means there is some kind of interaction between MTSL and BSA even if the label is free. The effect might be explained by collisional on-off contact of the free label with BSA. Because of these uncertainties we are careful not to draw speculative and far reaching conclusions.
- We were thinking to omit the MTSL data, however, the slower rotational dynamics in the BSA+TiO2 samples is interesting as it means either a direct effect of TiO2 on loosely bound label or effect on the free label (whichever the identity of the mobile component is). In addition, the shape of the spectra and the strong presence of the 5-peak component could be extremely useful for EPR spectroscopists working with MTSL on BSA or other protein.
- Regarding SASL binding indicating a partial unfolding of BSA, as pointed out above, we can exclude it. We thank the reviewer to raise the differences in SASL binding because we have just noted that we failed to clearly emphasise that the filtrated BSA samples have ~3.3-fold lower protein concentrations than the BSA and BSA+TiO2 samples. We have made it now clear.
- The data from MSL is now clear as well as the sample names.
- Regarding component separation in the MTSL spectra, briefly: we changed the MTSL concentration yielding varying 5-peak contribution. By making cross-subtractions we could identify the 5-peak component and remove it from the BSA spectra, ending up with a normal mobile component plus (a very small) immobile one. The algorithm for optimised subtraction and component separation are described in (Pali and Kota, 2019) in great detail (it is a tutorial-like chapter in a methodology book). It is true that the fractional contribution of the components are not interesting, and we are not discussing that. However, we had to separate them to be able to derive the rotational correlation times for the mobile component (which is interesting).
- Regarding labelling efficiency, since we did not remove free label (by washing), so the efficiency of covalent labelling is represented by the immobile fraction. This is meaningful in the case of 5-SASL and 5-MSL but very small for MTSL (and we don't know if the mobile MTSL component is free or loosely bound).
-> Changes:
We have added more details in section 2.5.3 (which was 2.4.3 in the previous revision) about how the subtractions were made on the 3-component MTSL spectra.

6.    "Ultimately, the revisions made after the initial review have steered the manuscript in a more favourable direction. However, substantial improvements are still required for the manuscript to be considered for publication."
-> Answer:
We thank the reviewer for improving our manuscript with important comments and requests.
-> Changes: We have corrected some typos and small errors, and made stylistic improvements. All changes are marked in the PDF version of the second revision of our manuscript.

Reviewer 2 Report

Despite the CD and detailed DLS/ZP measurements are not available for this study, the reviewer is sure that some findings (especially EPR related) should be delivered to the readers of the journal. The recommendation is accept. 

The quality of the English is acceptable.

Author Response

We thank the reviewer for the review with important suggestions and accepting our revised manuscript.

Round 3

Reviewer 1 Report

The manuscript's readability has improved, but there is still room to make it better. The authors should provide clear definitions for the microfiltration and ultrafiltration methods. Please maintain consistency in the units used, choosing either g/l or uM throughout. In the Results section concerning EPR, it is crucial for the authors to explicitly mention that they did not remove the excess spin probe from samples. This method deviates from the norm and should be explicitly stated, not only in the Methods section, as most readers would typically assume that the samples are free from excess spin label.

Can be improved.

Author Response

We thank the reviewer for valuable new suggestions to further improve our manuscript. Below are our point-by-point responses to their three requests:   1. "The authors should provide clear definitions for the microfiltration and ultrafiltration methods." -> Answer: The aim of ultrafiltration experiments was to investigate the effect of TiO2 coating on filtration performance of composite membranes, while microfiltration (0.22 um CA syringe filter) was used in another series of experiments aimed investigating the potential effect of TiO2 nanoparticles on BSA structure and filtration behaviour. In this case the aim of microfiltration was to separate TiO2 nanoparticles from BSA solution. -> Changes: We have made this clear by adding this explanation to the beginning of section 2.1. of the Results, and a new sentence to the 2nd paragraph of section 3.1 of the Methods.   2. "Please maintain consistency in the units used, choosing either g/l or uM throughout." -> Answer/Changes: We have converted all concentrations from g/l to M unit. We have also made minor corrections to the text in several places.   3. "In the Results section concerning EPR, it is crucial for the authors to explicitly mention that they did not remove the excess spin probe from samples. This method deviates from the norm and should be explicitly stated, not only in the Methods section, as most readers would typically assume that the samples are free from excess spin label." -> Answer/Changes: We have added two new sentences in section 2.3. stating the above detail, and we now also mention it in section 2.5.